# Ranking microbial metabolomic and genomic links in the NPLinker framework using complementary scoring functions

**Grímur Hjörleifsson Eldjárn**[1], **Andrew Ramsay**[1], **Justin J. J. van der Hooft**[2],
**Katherine R. Duncan**[3], **Sylvia Soldatou**[4], **Juho Rousu**[5], **Rónán Daly**[6],
**Joe Wandy**[6], **Simon Rogers**[1]*

**1** School of Computing Science, University of Glasgow, Glasgow, United Kingdom, **2** Bioinformatics Group, Wageningen University, Wageningen, The Netherlands, **3** Strathclyde Institute of Pharmacy and Biomedical Sciences, University of Strathclyde, Glasgow, United Kingdom, **4** School of Pharmacy and Life Sciences, Robert Gordon University, Aberdeen, United Kingdom, **5** Department of Computer Science, Aalto University, Espoo, Finland, **6** Glasgow Polyomics, University of Glasgow, Glasgow, United Kingdom

\* simon.rogers@glasgow.ac.uk

**Data Availability Statement:** All relevant data are within the manuscript and its Supporting information files.

## Abstract

Specialised metabolites from microbial sources are well-known for their wide range of bio-medical applications, particularly as antibiotics. When mining paired genomic and metabolomic data sets for novel specialised metabolites, establishing links between Biosynthetic Gene Clusters (BGCs) and metabolites represents a promising way of finding such novel chemistry. However, due to the lack of detailed biosynthetic knowledge for the majority of predicted BGCs, and the large number of possible combinations, this is not a simple task. This problem is becoming ever more pressing with the increased availability of paired omics data sets. Current tools are not effective at identifying valid links automatically, and manual verification is a considerable bottleneck in natural product research. We demonstrate that using multiple link-scoring functions together makes it easier to prioritise true links relative to others. Based on standardising a commonly used score, we introduce a new, more effective score, and introduce a novel score using an Input-Output Kernel Regression approach. Finally, we present NPLinker, a software framework to link genomic and metabolomic data. Results are verified using publicly available data sets that include validated links.

## Author summary

In this article, we introduce NPLinker, a software framework to link genomic and metabolomic data, to link microbial secondary metabolites to their producing genomic regions.

Two of the major approaches for such linking are analysis of the correlation between sets of strains, and analysis of predicted features of the molecules. While these methods are usually used separately, we demonstrate that they are in fact complementary, and show a way to combine them to improve their performance.

We begin by demonstrating a weakness in the most common method of strain correlation analysis, and suggest an improvement. We then introduce a new feature-based analysis method which, unlike most such methods, does not directly depend on the natural product compound class. Finally, we demonstrate that the two are complementary and

**Funding:** JJJvdH acknowledges an ASDI grant from the Netherlands eScience Center - NLeSC (grant no. ASDI.2017.030, https://www.esciencecenter.nl/). AR, KRD and SR acknowledge funding from the Biotechnology and Biological Sciences Research Council (BB/R022054/1, https://bbsrc.ukri.org/). KRD, SR and SS are supported by a Carnegie Trust Collaborative Research Grant (https://www.carnegie-trust.org/). JR acknowledges funding from the Academy of Finland (grants 310107 and 334790, https://www.aka.fi/) and Scottish Informatics and Computing Science Alliance (SICSA) distinguished visiting fellow scheme (https://www.sicsa.ac.uk/). The funders had no role in study design, data collection and analysis, decision to publish, or preparation of the manuscript.

**Competing interests:** The authors have declared that no competing interests exist.

proceed to combine them into a single scoring function for genomic and metabolomic links, which shows improved performance over either of the individual approaches.

Verification is done using curated databases of genomic and metabolomic data, as well as public data sets of microbial data including validated links.

# 1 Introduction

Microbial specialised metabolism, i.e. microbial production of metabolites not strictly needed for the survival of the organism, has been a rich source of metabolites for a variety of biomedical applications [1]. Recent advances in genomic analysis [2–4] indicate that microorganisms harbour considerable untapped metabolomic potential [5].

Genes responsible for the production of microbial specialised metabolites are usually grouped into Biosynthetic Gene Clusters (BGCs), contiguous regions of adjacent genes that, taken together, encode enzymes for the production of one or several structurally related metabolites. The most popular tool for BGC prediction in microbial genomes is antiSMASH [2], although several more exploratory tools exist (e.g. [4, 6]). Applying these predictive tools to novel genomes many new BGCs are predicted, the vast majority of which encode unknown products.

When searching for the products of BGCs, microbial extracts are often profiled using tandem mass spectrometry (MS/MS or MS2). Using this technique, an MS2 spectrum is recorded for a subset of the ionisable metabolites in a culture extract representing the fragmentation of the metabolite, providing useful information as to its structural identity. Data from multiple strains can be combined allowing researchers to see in which strains (or under which growth conditions) particular spectra appear.

Identifying from within these rich metabolomic datasets the particular molecular product (or products) of a BGC is a challenging, but important, problem. Firstly, since rediscovery of known metabolites is a persistent problem in metabolomics [7], linking a spectrum to a BGC with a known metabolite product can help identify the spectrum as belonging to that metabolite, and thus complement database matching as a dereplication strategy. An example of this approach can be seen in the characterisation of a polyketide antibiotic with an elemental formula of $C_{35}H_{56}O_{13}$ [8]. Secondly, since looking for metabolites similar to a known metabolite is a common starting point in microbial metabolomics (e.g. [9]), establishing the spectrum corresponding to the product of a BGC that is similar to a BGC with a known metabolite product, can be considered particularly useful. This strategy, for example, was used to isolate the novel lanthipeptide tikitericin from a thermophilic bacterium [10]. Lastly, knowing which BGC produces a particular metabolite can give valuable insight for synthetic biology applications, e.g. heterologous expression. This approach was used to identify the normally cryptic BGC responsible for the production of scleric acid, a secondary metabolite with moderate antimicrobial and antitumor activity [11].

Linking BGCs to their metabolite products has largely been done manually and on a small scale, working with single (or small numbers of) strains, either based on similarity to known, existing links (e.g. [10]), or predictions of unique identifying features of the spectrum from the BGCs (e.g. [12]). However, to increase the chance of success, this is increasingly being done at a large scale through the simultaneous analysis of a large number of related strains. The increased throughput makes manual methods prohibitively labour intensive, making development of computational methods for this purpose necessary.

To date, relatively few computational tools exist to aid in this BGC-MS2 linking problem. The problem is challenging: given a collection of BGCs in a strain, and a collection of metabolites produced by the same strain, any given metabolite could, a priori, be produced by any of the BGCs, yielding a huge number of potential links. To link BGCs to MS2 spectra, computational approaches have therefore concentrated on scoring this collection of all possible links, and then ranking the links by score. Such approaches can prioritise links for further analysis, in order to accelerate the process of finding the correct links in the set of all possible links. This can be both theoretical (by analysing the BGC to predict the properties of the product) and experimental (e.g. heterologous expression) [13–15]. Existing approaches to compute scores can be broadly classified into two categories: *feature-based approaches*, where the set of MS2 spectra is searched for predicted structural features of the putative product of the BGC (e.g. *peptidogenomics*, [16]), and *correlation-based approaches*, where similarities in sets of strains containing specific BGCs on one hand, and specific spectra on the other hand, are used to evaluate the links between BGCs and spectra (*metabologenomics*, [15, 17]).

A few tools exist to aid in *feature-based linking*, such as SANDPUMA [18], GNP [19] and MetaMiner [20]. These tools are each designed to be used for only a single product type such as nonribosomal peptides (NRPs), polyketides (PKs) or ribosomally synthesized and post-translationally modified peptides (RiPPs), respectively. Further analysis is therefore required to evaluate the links from different tools against each other.

The most popular *correlation-based* approach is described in [17]. Here, BGCs from different strains are clustered into Gene Cluster Families (GCFs) based on similarity-based distances between BGCs [13, 15] with the assumption that BGCs that are close with regards to this distance will produce identical or structurally similar metabolites. The current state-of-the-art tools for microbial BGC clustering are BiG-SCAPE [13] and BiG-SLICE [21]. Of the two, BiG-SCAPE offers higher precision, and as the data sets under consideration here are not of the size that performance becomes a problem, BiG-SCAPE is used for BGC clustering throughout this article. GCF and spectrum pairs are then scored based upon their shared strains. This approach was for instance used in [15] to link tambromycin to its producing BGC. As well as scoring links between GCFs and spectra, this approach can also be used to score links between GCFs and Molecular Families (MFs) produced by spectral clustering pipelines, such as the *molecular networking* pipeline within GNPS [22].

Fig 1 is a diagram showing the relationship between the various terms discussed above. For detailed reviews of linking microbial specialised metabolites to their producing BGCs, the reader is referred to [23] and [24]. Note that links can be defined between different entities. For example, on the genomic side, one can consider links involving individual BGCs (feature based linking) or GCFs (feature based or correlation based linking). On the metabolomics side, one can use MFs or individual spectra for either feature- or correlation-based linking (both of which have been successfully used, e.g. [9] and [15], respectively).

In this work, we address various limitations of the currently available computational approaches.

- Firstly, as we will show, the most popular strain correlation score [17] has properties that make it impossible to reliably compare score values across links (even links from the same GCF or MF), severely limiting the scores it produces for ranking links. We propose a method for standardising this score such that link scores become comparable across a whole dataset, or for a particular GCF/MF.

- Secondly, although feature-based methods exist for particular specialised metabolite classes (e.g. [18, 19, 25]), there is no general feature-based method for ranking links across all

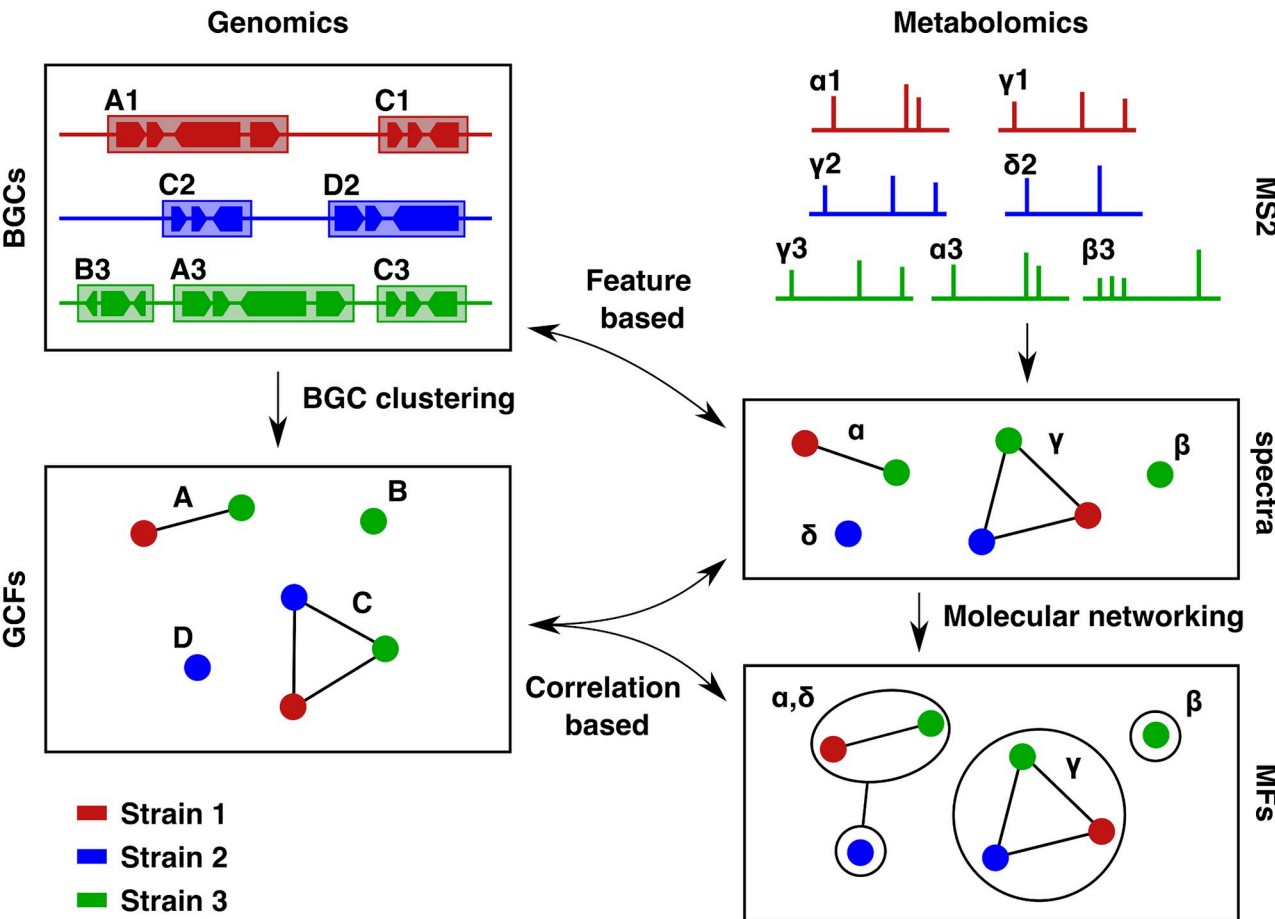

**Fig 1. Diagram showing the relationship between the various metabolomic and genomic objects.** On the genomics side, BGCs are detected from microbial genomes, colour-coded by strain. These are clustered into GCFs, where each GCF contains BGCs from one or more strains. GCFs can thus also be considered as sets of strains, where each strain contributes at least one BGC to the GCF. On the metabolomics side, MS2 spectra measured in microbial cultures are grouped across strains, so that identical spectra are assigned one or more strains in which they appear. These are further grouped into MFs in a process called Molecular Networking, where each MF consists of one or more related spectra. Both spectra and MFs can likewise be considered as sets of strains where the spectrum, or a spectrum in the MF, is present in the sample for the strain. Feature-based approaches can be used to link BGCs to individual spectra, while correlation-based approaches can be used to link GCFs to either MFs or spectra, based on the pattern of strain contents.

classes. Inspired by progress in the area of *in silico* metabolite identification, in particular [26], we propose the use of Input-Output Kernel Regression (IOKR) for this task.

- Thirdly, no previous study has attempted to combine feature- and correlation-based scores, despite the fact that they are likely to be complementary. Here we show that strain correlation and IOKR scores are indeed complementary and present a method for combining their scores into a single value.

- Finally, we introduce an open source extendable software framework—NPLinker—into which our developed methods have been implemented that makes it straightforward to import the various data types and will allow accelerated development of new methods for this problem.

## 2 Materials and methods

### 2.1 Strain correlation scoring

Consider a population of strains, each with a set of predicted BGCs. The union of those sets constitutes a set of BGCs in the population. Assume also that the BGCs have been clustered into GCFs. Similarly, assume that associated with the population is a set of all MS2 spectra for metabolites produced by the population, clustered into MFs. We are interested in scoring these potential GCF-MF links, in order of how likely each GCF is to produce the metabolite giving rise to each MF.

The most common scoring method for scoring a GCF-MF link in metabologenomics is as follows [17]: starting from zero, for each strain in the population,

- add 10 to the score if the strain produces the metabolite and has a BGC in the GCF,

- subtract 10 from the score if the strain produces the metabolite but does not have a BGC in the GCF,

- add 1 to the score if the strain neither produces the metabolite nor has a BGC in the GCF, and

- leave the score unchanged if the strain has a BGC in the GCF but does not produce the metabolite.

Considering the GCF as a set $G$ of strains that contribute a BGC to the GCF, and the MF as a set $M$ of strains that produce a metabolite in the MF, the score is therefore defined as a function $\sigma_{\mathrm{corr}}(M, G)$ which depends on the size of $M$ (#$M$), the size of $G$ (#$G$), and the overlap between the two sets, in addition to a set of weights (here 10, -10, 1 and 0, respectively).

While intuitive and easy to interpret, this scoring function, hereafter referred to as *strain correlation score*, has the major drawback of being highly dependent on both total population size and, with potentially greater impact, the size of the GCF and the number of strains that produce the metabolite. A link between a GCF that contains a BGC from every strain in the data set and a moderately-sized MF can easily outscore a link between a smaller GCF and MF that have perfect strain correspondence, even if the evidence for the latter link being valid is stronger, as shown in Fig 2(A). This makes comparison between possible links involving GCFs and MFs of different sizes challenging, even within the same data set.

### 2.2 Standardisation of the strain correlation score

This effect of GCF and MF sizes on the strain correlation score can be mitigated by standardising the score. For a given GCF and MF, let $G$ and $M$ be the sets of strains contributing to the GCF and the MF respectively. Assuming the null hypothesis that the strain content of $G$ and $M$ is not correlated, the probability of a given overlap between the two follows a hypergeometric distribution with the total number of strains as the population size $N$, the size of the molecular family $M$ as the number of positives in the population, the size of the GCF $G$ as the sample size, and the size of the overlap between the two as the number of positives in the sample.

Letting $m = \#M$, $g = \#G$, $o = \#(G \cap M)$ and $n = \#N$, we can define the strain correlation $\sigma_{\mathrm{corr}}(M, G) = \sigma'_{\mathrm{corr}}(m, g, o, n)$ in terms of the sizes of $M$, $G$ and the overlap between the two. We can therefore compute the expected value $E[\sigma_{\mathrm{corr}}(M, G)]$ as a function of the sizes of $M$ and $G$ as

$$E[\sigma_{\mathrm{corr}}(M, G)] = \sum_{k} p(o = k)\sigma'_{\mathrm{corr}}(m, g, k, n)$$

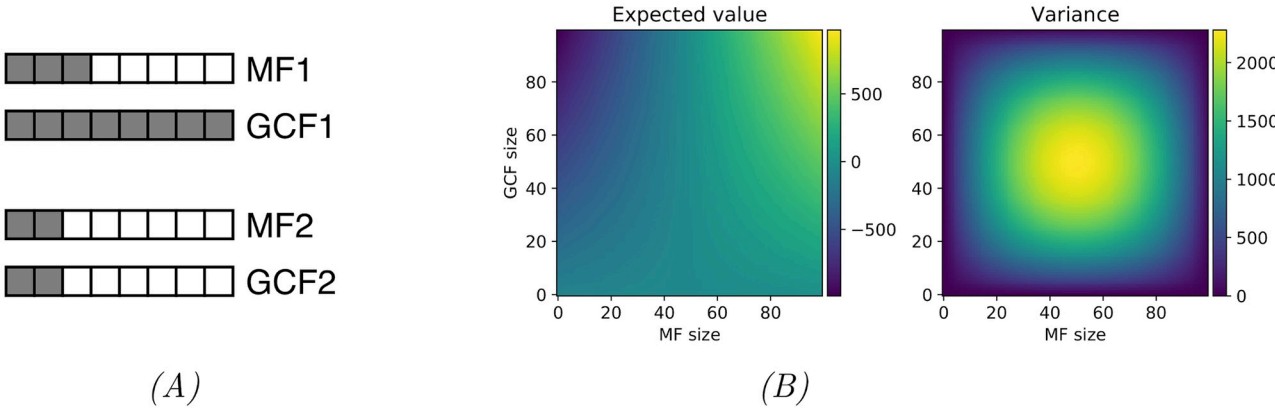

**Fig 2. The effect of size on strain correlation scoring.** *(A)* Size discrepancy in the strain correlation score for GCFs of varying sizes. Each box represents a strain, with filled boxes denoting that the strain is a member of the GCF or MF, and blank boxes that it is not. The top GCF-MF pair outscores the bottom pair by 30 to 26, despite the bottom pair having arguably stronger correspondence. *(B)* Expected value and variance of the strain correlation score for a population of 100 strains, as a function of GCF and MF sizes. Both the expected value and the variance have a considerable range, rendering comparison between links involving different sizes of GCFs and MFs difficult. For instance, a GCF and MF of size 80 could easily get a score of 500 or higher by chance, while for a GCF and MF of size 20, a score this high would be highly significant.

where $k$ runs over all possible sizes of $(M \cap G)$, and $p(o = k)$, the probability of the size of the overlap $\#(M \cap G)$ being $k$, follows the hypergeometric distribution as previously stated. The variance can then be computed as

$$Var[\sigma_{corr}(M, G)] = E[\sigma_{corr}(M, G)^2] - E[\sigma_{corr}(M, G)]^2$$

Fig 2(B) shows the expected value and variance of the strain correlation score as a function of the sizes of $G$ and $M$, for population size $\#N = 100$. The expected value varies greatly, especially with the number of strains producing the metabolite (due to the higher magnitude coefficients when the spectrum is present in the strain than when it is absent), and the variation in variance across the domain is also considerable.

We propose to mitigate this by defining a *standardised strain correlation score* for a GCF $G$ and MF $M$ as

$$\bar{\sigma}_{corr}(M, G) = \frac{\sigma_{corr}(M, G) - E[\sigma_{corr}(M, G)]}{Var[\sigma_{corr}(M, G)]}$$

In this version, the score for each prospective link takes into account the sizes of the GCF and the MF and adjusts the score accordingly, making the scores comparable between links involving strain sets of different sizes as is necessary when, for example, comparing scores for different spectra or MF for a particular GCF. While still not having an upper or lower bound, the standardised correlation score has mean 0, and variance 1. In the case of Fig 2(A), the standardised scores are 0.0 and 2.65, favoring the bottom pair.

A standardised strain correlation score of zero means that the degree of overlap between the two strain sets is the same as would be expected if they were chosen at random, given the size of the GCF and the MF. For instance, since the GCF in the example contains all the strains in the population, any MF will have complete overlap with the GCF, giving all links for this GCF a score of zero. Calculation of significance for the strain correlation score can be found in Appendix A in S1 Text.

## 2.3 Input-Output Kernel Regression

If a BGC is known to produce a certain metabolite, the problem of linking a spectrum to that BGC is equivalent to linking a spectrum to the metabolite that the BGC produces. This problem, of matching spectra to molecular structures, is an important problem in metabolomics because it underpins all untargeted metabolomics workflows. This matching is often done in the space of *molecular fingerprints*, which are binary vectors denoting the various properties of the molecule, including the presence or absence of certain substructures. These fingerprints can be derived from candidate structures and effectively predicted from spectra.

Brouard *et al.* propose Input-Output Kernel Regression (IOKR) [26, 27] as a method of ranking a candidate set of chemical structures, given an input spectrum. They compare IOKR with state of the art methods such as CSI:FingerId [28] and demonstrate similar performance with considerably shorter training and classification times, with the training time reduced from 82 hours to under one minute, and classification time reduced by half.

In principle, IOKR works by first learning a mapping from the space of spectra to the space of molecular fingerprints. This mapping, along with the function mapping molecular structures to their molecular fingerprints, is then used to project similarities in the input space (spectra) and the output space (molecular structures) to the molecular fingerprint space. This allows searching a candidate set of metabolites for the element giving the closest match in the space of molecular fingerprints to the predicted fingerprint for the spectrum. Let $\mathcal{X}$ be the space of MS2 spectra and $\mathcal{Y}$ the space of metabolites, with kernel functions $\mathcal{K}_x, \mathcal{K}_y$, respectively, and $X \subset \mathcal{X}$ and $Y \subset \mathcal{Y}$ be training sets of paired spectra and metabolites, where each $x_i \in X$ has a corresponding element $y_i \in Y$, and vice versa. Let $\mathcal{F}$ be the space of molecular fingerprints, and $\hat{h} : \mathcal{X} \to \mathcal{F}, x \mapsto \sum_{x_j \in X} \mathcal{K}_x(x, x_j) \mathbf{c}_j$, where $\mathbf{c}_j \in \mathcal{F}$ are vectors in the fingerprint space. Finally, let $\phi : \mathcal{Y} \to \mathcal{F}$ be the mapping expressing the kernel function $\mathcal{K}_y$ as the inner product in $\mathcal{F}$, $\mathcal{K}_y(\cdot, \cdot) = \langle \phi(\cdot), \phi(\cdot) \rangle_{\mathcal{F}}$. Given a spectrum $x$, IOKR works by searching a set $Y^*$ of candidate structures for an element $y$ that maximises the expression $\langle \hat{h}(x), \phi(y) \rangle_{\mathcal{F}}$. Fig 3 is an arrow diagram of the IOKR framework.

For a BGC $g$ with an associated molecule $g'$ and spectrum $m$, we can then define a link scoring function $\sigma_{\mathrm{IOKR}}$ as

$$\sigma_{\mathrm{IOKR}}(m, g) = \langle \hat{h}(m), \phi(g') \rangle_{\mathcal{F}}$$

For further details of the underlying mathematics of IOKR, please refer to [27].

## 2.4 Using IOKR to rank BGC-spectrum links

In the context of linking BGCs and spectra, IOKR can be considered a *feature-based method*. When scoring links between individual BGCs and spectra, however, it does not predict the features of the spectrum directly from the BGC, but uses molecular fingerprints as an intermediate. These can be predicted from the MS2 spectra, using a function learned from annotated training data, and calculated from structural predictions for the BGCs. Notably, IOKR is not limited to particular product classes, as it requires only spectra and predicted molecular structures.

The application of IOKR depends heavily on the choice of kernel function on the spectra, and on the choice of molecular fingerprint. For this work, we choose kernel functions that are easy and fast to compute, with the caveat that further optimisation may be possible.

As stated in Section 2.3, IOKR works by ranking a set of candidate structures by similarity to a given spectrum. Therefore, this method is not directly applicable to BGCs unless the chemical structure of the metabolite produced by the BGC is known. Generally, this is not the

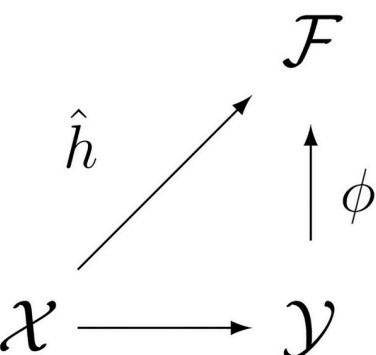

**Fig 3. Arrow diagram of the Input-Output Kernel Regression (IOKR) framework.** $X$ denotes the space of MS2 spectra, $\mathcal{Y}$ is the space of metabolites, and $\mathcal{F}$ is the shared space of molecular fingerprints. $\hat{h}$ is the (learned) mapping from MS2 to fingerprints, while $\phi$ is the (exact) mapping from metabolites to molecular fingerprints.

case. However, by considering only the BGCs which have considerable similarity to BGCs in annotated libraries such as MIBiG, structural predictions can be made for a subset of all predicted BGCs. Potential links between spectra and BGCs belonging to this subset can then be ranked using IOKR.

Molecular fingerprints are extracted from SMILES strings using the Chemistry Development Kit [29]. The fingerprint vector is composed of three concatenated sets of fingerprints: *CDK Substructure*, *PubChem Substructure* and *Klekota-Roth* fingerprints. Taken together, these cover most of the molecular properties described by the fingerprint used by Brouard *et al.* and result in similar performance [26].

As a denoising step, to avoid time-consuming computation of fragmentation trees for the spectra, we filter the input spectra to include only the peaks found in the training data, before using the *Probability Product Kernel* (PPK) [26, 30]. While this in theory might bias the model towards the training set, in practice this approach is widely used. The training set is large enough that it contains enough ions to build a robust model, and the effect of the bias is small, as borne out by comparing this filtering approach to the filtering approach used by PPKr in [27], which filters peaks based on proposed fragmentation trees.

Since the standardised strain correlation score is defined between MFs and GCFs, and the IOKR score between BGCs and spectra, they are not directly comparable. To be able to use them together, we generalise the IOKR score to GCF-MF links by taking the highest-scoring BGC-spectrum pair where the BGC is in the GCF and the spectrum in the MF, and assign that score to the GCF-MF pair, i.e. for a GCF $G$ and a MF $M$, and a scoring function $\sigma_{\text{IOKR}}$ scoring BGC-spectrum links, we define a second function with the sets of GCFs and MFs as a domain, by setting $\sigma_{\text{IOKR}}(M, G) = max_{m\in M, g\in G}\sigma_{\text{IOKR}}(m, g)$.

## 2.5 Combining strain correlation and feature-based scores

So far, computational efforts to find GCF-MF links have mostly been based on either the *feature-based* or the *correlation-based* approach. However, both approaches have limitations. For example, correlation-based methods have trouble prioritising singleton links within the same strain, i.e. when the same strain is associated with multiple singleton GCFs and MFs. Feature-based methods on the other hand rely on being able to predict distinguishing features that can be detected in MS2 spectra from the BGCs, and since the same features are often present in multiple GCFs and MFs, this can yield multiple potential links with the same score.

Because the two approaches discussed in this article are based on very different principles, they are likely to be at least in part complementary, with each one sensitive to things that the other is not. Since both approaches work by considering the set of all possible GCF-MF combinations in the data set and assigning scores to each one, we can consider the scores for the links as assigning to each link a point in $\mathbb{R}^2$, given by $(\bar{\sigma}_{\text{corr}}, \sigma_{\text{IOKR}})$. We demonstrate in Section 3.4 that combining the two approaches by looking at potential links that have a robust ranking using both scoring functions increases the ratio of previously validated links to all links in the joint top percentiles compared to using either one of the approaches. Furthermore, Section 3.5 introduces concrete ways of combining the scores to make use of this complementarity.

## 2.6 NPLinker

To facilitate analysis of paired genomics and metabolomics data sets, we developed NPLinker, a Python module to accelerate and support the process of automatically linking GCFs or their BGCs with observed mass spectra. Demonstrating its utility, NPLinker has been used by Soldatou and co-workers to putatively link the known metabolites ectoine and chloramphenicol to their producing BGCs [31].

NPLinker accepts genomic outputs from antiSMASH and BiG-SCAPE (including reference BGCs from the MIBiG database [32]), and metabolomic output from the public, community-driven Global Natural Products Social (GNPS) knowledge base [33]. Additionally, it includes integration with the *Paired omics Data Platform* [34] to retrieve paired public genomics and metabolomics data (https://pairedomicsdata.bioinformatics.nl).

In addition to the framework, NPLinker includes a user interface for visual inspection of the potential links, and can be used as a stand-alone web app. For ease of use, it can be run in a Docker container, working with either local data, links to data sets in the Paired omics Data Platform or a mixture of both.

After loading the metabolomic and genomic data, including automated running of BiG-SCAPE if required, links can be sorted, inspected and filtered by various scoring functions or combinations thereof, and visualised as tables. A screen shot and further documentation for NPLinker are included in Appendices B and C in S1 Text.

NPLinker creates objects for spectra, MFs, BGCs and GCFs in the data set, maintaining the hierarchical relationship between them, and keeps track of strain ID or IDs associated with each object, as well as strain aliases. Objects can also be filtered by various criteria such as strain association, inclusion in MIBiG, and annotations.

NPLinker creates a set of hypothetical links between the metabolomic and genomic objects, which can then be evaluated using various scoring functions, both scoring functions that are supplied with NPLinker and custom scoring functions. Not all scoring functions need to be defined for all potential links. For instance, in its current formulation, the IOKR scoring function described here can only rank links involving BGCs that have predicted molecular structures, which in practice means BGCs with significant homology to known BGCs, whereas the strain correlation score can rank any GCF-MF link. Other scoring functions may for instance only apply to particular product types. Fig 4 shows how NPLinker combines the various data sources.

The NPLinker source code and documentation can be found at https://github.com/sdrogers/nplinker, while the version used for the analysis in this work can be found at http://doi.org/10.5281/zenodo.4680579.

## 2.7 IOKR training data

The IOKR model is trained on a set of spectrum-molecular fingerprint pairs. Since molecular fingerprints can be computed from structural annotations, such as SMILES [35] or InChI [36],

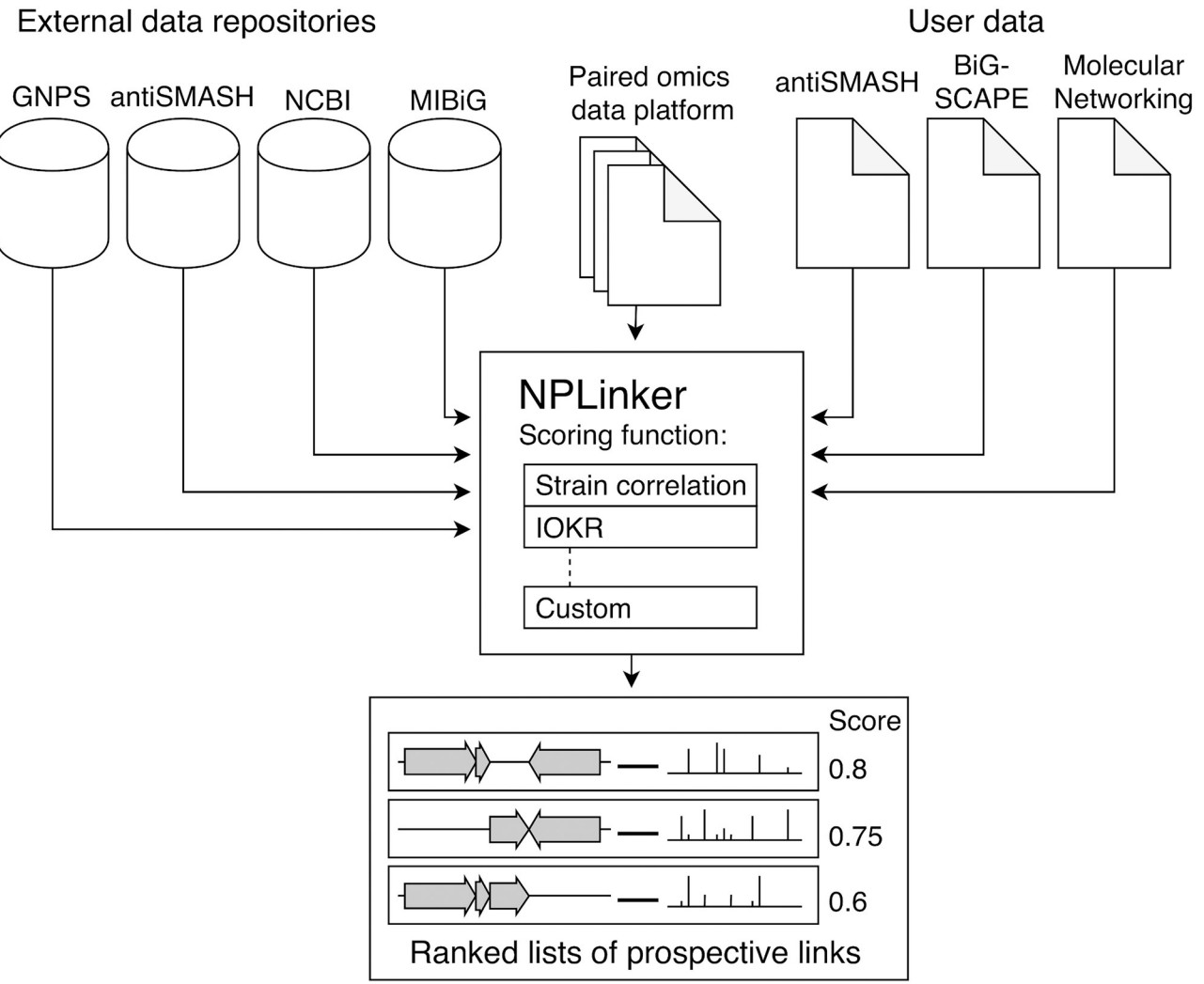

**Fig 4. Diagram of the NPLinker module.** The NPLinker module helps with automatically linking GCFs and MFs. It integrates metabolomic and genomic data sets, using either external sources, user-provided data, or a mixture of both, and ranks potential links between metabolomic and genomic objects by given scoring functions, either built-in or user-defined.

we use library MS2 spectra from the public, community-driven GNPS knowledge base [33] as a training set for the IOKR model. We use the same training data set as Brouard and co-workers [26], which consists of 4138 spectra from GNPS with structural annotations. The spectra are for metabolites from a variety of sources, including microbial, plant and human metabolites. While GNPS contains a far higher number of spectra with structural annotations, this subset represents high-quality MS2 data with similar device configuration and thoroughly verified structural annotations. Furthermore, this same subset has been used in numerous previous articles (such as [27] and [28]), making performance comparisons between different models easier.

## 2.8 Test data

While NPLinker can in principle accept BGCs and GCFs from any source as input, at present it is aimed towards the BiG-SCAPE workflow, in order to make use of strain correlation

scoring. As DeepBGC is not yet incorporated into the BiG-SCAPE workflow, evaluation on microbial data sets is carried out using antiSMASH-predicted BGCs, both in the case of the standardised strain correlation, where the GCF information is required, but also in the case of IOKR, where only the BGC information is required. In practice, this is not likely to present much of a problem: as the main improvement of DeepBGC on antiSMASH lies in improved detection of novel BGCs, and to evaluate IOKR, we must restrict our analysis to BGCs for which reliable structural predictions exist—i.e. BGCs with strong homology to MIBiG BGCs— the difference between antiSMASH and DeepBGC is likely to be very slight for the BGCs in the test set.

Separate validation on IOKR, on a data set not dependent on antiSMASH, is carried out on a data set created by matching manually curated BGCs from MIBiG [32] to manually curated spectra from GNPS [22] using structural annotations from the databases.

**2.8.1 Pairing the MIBiG and GNPS databases.** In recent years, the MIBiG database [32] has emerged as a central repository of characterised microbial BGCs. The current version, 2.0 [37], contains close to 2000 BGCs, most of which have structural annotations. Many tools, including antiSMASH and BiG-SCAPE, use MIBiG as part of their analysis to quantify similarity of unknown to known BGCs.

In particular, antiSMASH can be configured to compute the similarity of detected BGCs to MIBiG entries and return for each detected BGC a (ranked) list of similar BGCs in MIBiG, using the *known cluster blast* feature [2]. Assuming that similar BGCs give rise to similar compounds, we used this list in turn to assign one or more molecular structures to BGCs, according to how many high-scoring matches are found in MIBiG (or none, if no match was found).

While small changes in domain composition can potentially result in a big change in molecular structure, the fundamental premise of clustering BGCs into GCFs rests on the assumption that the distance between the BGCs is defined in such a way that the assumption above is true, at least for the majority of cases. However, verifying the validity of this assumption, and therefore the validity of the distance functions, is outside the scope of this paper.

In its current form, MIBiG has no information about the MS2 spectra of the metabolites produced by the BGCs. However, the entries are annotated with one or more structures, so the structural annotations included in GNPS can be used to link GNPS spectra to their corresponding MIBiG BGCs. In this way, we built a set of known BGC-spectrum pairs. To avoid distinguishing between metabolites based on properties absent from an MS2 spectrum, e.g. the chirality of the metabolite, this linking is done using only the first part of the InChIKey of each metabolite. This yields 2966 BGC-spectrum pairs, each with an associated metabolite, which can be used to evaluate the IOKR model proposed in this paper. These pairs include 2069 unique spectra and 242 unique MIBiG BGCs. The matched MIBiG and GNPS entries can be found in S1 Data. As MIBiG can contain more than one BGC encoding a given metabolite, the relation between spectra and BGCs in this list is many-to-many: some MIBiG BGCs can encode multiple metabolites, each of which can have multiple spectra in GNPS. Conversely, more than one MIBiG BGCs may encode the same metabolite, or different BGCs may enocde metabolites that have identical first parts of the InChIKey, and are therefore linked with the same spectra.

**2.8.2 Validated links from published data.** A major problem in the development of methods to link BGCs to spectra for collections of microbial strains is the lack of ground truth data. Given a data set, consisting of a collection of GCFs for a population of strains, and a collection of MFs for the same strains, the number of potential GCF-MF links is vast. Because microbial secondary metabolism is largely controlled by BGCs, many of these potential links will be true, in that the BGCs in the GCF are responsible for the production of the molecules in the MF. Only a very small number of these true links have been validated, however. Any

effective scoring function will therefore have a large number of true but unvalidated links towards the top end of the distribution.

The performance of a scoring function can be measured by the ratio of true links to all links at the top end of the distribution. The ideal scoring function would assign all true links a higher score than all false links, and as any improvement in ranking should increase the portion of true links and decrease the portion of false links towards the top of the distribution, comparing two scoring functions can be done by considering this ratio for both functions. While the set of validated links for each data set constitutes only a very small subset of the actual links, the same principle can be used to compare scoring functions using only the subset of validated links, i.e. comparing the ratio of *validated* links to all links, instead of the ratio of *true* links to all links, as increasing the ratio of true links to all links would in particular increase the ratio of validated links to all links.

Recognising the difficulty of obtaining ground truth data in this field, Schorn *et al.* recently developed the Paired Omics Data Platform documenting the location of genomic and metabolomic data sets from microbial experiments, with a focus on data sets with BGCs and MS2 spectra [34]. This gives a repository of validated links in various data sets, citing the articles in which the links were validated, which can then be used to evaluate scoring methods for prospective links between BGCs and spectra, in terms of relative over-representation of validated links towards the upper end of the distribution of scores. Although the platform contains a large number of data set with validated links, the minority of these contains detailed links between particular BGCs and MS2 spectra, and fewer still of these contain a large enough number of strains for the strain correlation score in particular to carry sufficient information. From this platform we concentrated on three data sets each with numerous validated links: MSV000078836 [38], MSV000085018 [39] and MSV000085038 [40], hereafter referred to as Crüsemann, Gross and Leão, respectively.

The Crüsemann data set consists of 120 microbial strains with 8 validated links between a BGC and a MF, the Gross data set consists of 7 strains with 9 validated links between a BGC and a MF, and the Leão data set contains 4 strains with 5 validated links between a BGC and a specific MS2 spectrum. After downloading the strain assemblies and metabolomics data, the genomes were run through antiSMASH v5.0.0 for BGC detection and BiG-SCAPE v1.0.0 to cluster the BGCs into GCFs. The validated links are of various product types, although primarily polyketides and NRPs, with sizes ranging from 7000 to 20000 nucleotides, or between four and 102 genes.

The Paired Omics Data Platform (PoDP) links are represented by molecular families or MS2 spectra in specific GNPS data sets, and by MIBiG IDs in specific strains. To map the BGC links back to the genomes in question we used antiSMASH to score the correspondence between the MIBiG entries and the detected BGCs, returning for each BGC a (possibly empty) list of MIBiG matches, along with a score for each match. To avoid excessive noise due to low-quality matches, the cutoff for this matching was a cumulative BLAST score of 10000, but the data sets being analysed did not appear very sensitive to variation in this value. This means that a validated link from the PoDP can link a single spectrum to multiple GCFs, all of which we consider as validated. This can indicate potential splitting of a cluster of similar BGCs, or ambiguity in product type for the BGCs. A full list of BGCs in validated links, their product types and sizes is in Table A in S1 Text. The sizes of the data sets can be found in Table B in S1 Text.

When clustering BGCs, BiG-SCAPE clusters the BGCs separately by product type. BGCs belonging to the class "PKS-NRP Hybrid" are treated as belonging to three classes: PKS, NRPS and Hybrid, and are therefore clustered three times. After clustering, identical GCFs are removed, so if identical clusters appear in for instance PKS and NRPS, that cluster only

appears once in the resulting data. The final outcome is that a hybrid BGC can belong to one, two or three GCFs.

For instance, considering the Crüsemann data set at the GCF-MF level gives 10 validated links for the MIBiG ID BGC0000137. That BGC is linked to two MFs in the data set, and two BGCs from the relevant strain show significant homology to the MIBiG BGC. Because both of the BGCs are assigned the product class "PKS-NRP Hybrid", BiG-SCAPE considers them as potential members of a PKS cluster, NRPS cluster or a hybrid cluster. This yields two potential clusters for one of the BGCs and three for the other BGC, giving a total of five potential GCFs linked to the two MFs, and therefore ten individual GCF-MS2 links.

## 3 Results

### 3.1 Standardising the strain correlation score

As the Crüsemann data set contained the largest number of strains of the three data sets being considered, it was selected to evaluate the standardised strain correlation score. To do this, we examine the distribution of scores for validated links in relation to the scores for all hypothetical links. The more effective a scoring method is, the better rank it should assign to the correct GCF-MF links compared to the rest of the links. Therefore, the mean score for validated links should be higher than the mean score for all potential links, so the null hypothesis for testing the validity of the scoring function is that both distributions of scores (for validated links and all links) have the same mean.

For the raw strain correlation score, the mean score is 83.514 for all links, and 14.667 for validated links, i.e. a lower mean score for the validated links than for all links, which is the opposite of what would be expected. Standardising the score gives a mean score of -0.006 for all links, and 3.672 for validated links (Table 1). The distributions of the scores of the validated links amongst all links are shown in Fig 5 for both the standardised and raw versions of the score. Similar trends can be observed in the Leão and Gross data sets, see Table C in S1 Text.

Table 2 shows the proportion of validated links among all links in the microbial data sets, both for all the potential links (first row), and for the links scoring above the 90th percentile using the various scoring functions. In particular, rows two and three show the proportion of validated links among the top scoring links for the raw and standardised correlation scores. In the Crüsemann data set, where the strain correlation scoring is most relevant, the proportion is considerably higher for the standardised correlation score relative to the raw correlation score.

The results demonstrate the importance of our proposed standardisation process.

### 3.2 Evaluating the IOKR scoring function using MIBiG

As stated earlier, the training set used to build the IOKR model includes metabolites from sources other than microbial. As the performance evaluation of IOKR does not break down

**Table 1. Mean scores for all links and the subset of validated links in the Crüsemann data set.**

| Method | all | validated | p-value |
|---|---|---|---|
| Raw correlation | 83.5144 | 14.6667 | 0.0001 |
| Standardised correlation | -0.0060 | 3.6717 | $6.8302 \times 10^{-64}$ |
| IOKR | 0.0105 | 0.0364 | $1.7968 \times 10^{-9}$ |

The mean score for validated links using the raw strain correlation score is in fact lower than the mean for all potential links, while for both standardised strain correlation score and IOKR, the mean for validated links is higher than the mean for all potential links. Note that the scores are not guaranteed to be on the same scale, so they are not directly comparable.

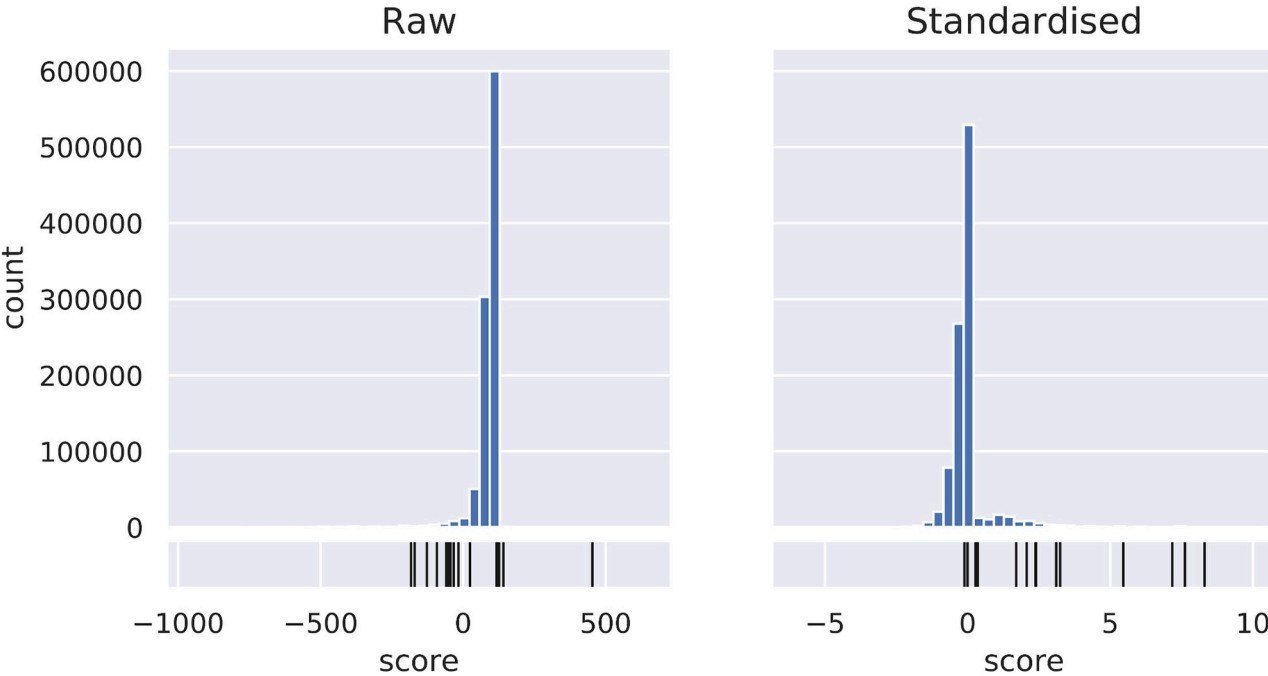

**Fig 5. Distribution of validated links among scores.** Distribution of the raw and standardised strain correlation scores, as well as the distribution of the scores for validated links (in black) relative to the distribution of scores for all links, in the Crüsemann data set. The standardised score has a more pronounced tail at the top end, which includes 13 out of 15 validated links, whereas many of the validated links score relatively low on the distribution of the raw scores. Figures for other data sets can be found in S1 Fig.

performance by metabolite sources, to evaluate the performance of IOKR on microbial specialised metabolites specifically, we tested the method on the paired MIBiG/GNPS data by matching each spectrum to the candidate set consisting of all structures associated with an MIBiG entry. For each spectrum, IOKR returns an ordered list of all metabolites in the candidate set. As multiple BGCs can produce the same metabolite, and as the same BGC can be associated with a number of structurally related metabolites, this can result in multiple BGC-MS2 links for the same metabolite.

The MIBiG/GNPS data set consists of sets of associated BGC, metabolite and spectrum. As implemented in [27], IOKR gives a ranking of links between spectra and metabolites, in our case using as a candidate set all metabolites that occur in the data set. To find the ranking of the correct metabolite match for a given spectrum, we take the rank of the best-ranking metabolite that is associated with the spectrum. To turn that ranking into the ranking of the correct BGC for the spectrum, we take the set of metabolites that are assigned better ranking than that

**Table 2. Proportion of validated links among all possible GCF-MF links in the three data sets.**

| Data set | Crüsemann | Gross | Leão |
|---|---|---|---|
| Total | $1.50 \times 10^{-5}$ | $9.96 \times 10^{-6}$ | 0.00086 |
| Top raw corr. | $6.22 \times 10^{-5}$ | $3.48 \times 10^{-5}$ | 0.00266 |
| Top std. corr. | 0.00013 | $7.69 \times 10^{-5}$ | 0.00385 |
| Top IOKR | $6.00 \times 10^{-5}$ | $1.99 \times 10^{-5}$ | 0.00107 |
| Top comb. | **0.00046** | **0.00019** | **0.00680** |

**Table 3. Top-*n* accuracy, and AUC, of IOKR on MIBiG data.**

|          | top-1  | top-5  | top-10 | top-20 | top-200 | AUC    |
|----------|--------|--------|--------|--------|---------|--------|
| data     | 0.1208 | 0.1708 | 0.1870 | 0.2121 | 0.2946  | 0.6534 |
| random   | 0.0    | 0.0014 | 0.0044 | 0.0103 | 0.1486  | 0.5209 |

metabolite, and the set of BGCs associated with any of those metabolites. The rank of the correct BGC is the number of BGCs that are associated with a metabolite that is assigned better rank than the best-ranking metabolite associated with the spectrum.

For comparison purposes, a baseline score was estimated by randomising the rank of the structures for each spectrum, and the same process was repeated to assign a BGC using the randomised score.

Table 3 shows the top-*n* performance of IOKR, i.e. how often the 'true' BGC match for a given spectrum is among the top *n* matches returned by IOKR, for a selection of *n*. Matching spectra to a set of candidate molecules is a very hard problem, and achieving top-1 accuracy above 20% is highly non-trivial, unless the candidate set is heavily restricted. For instance, the single best-performing kernel used by Brouard and co-workers in [27] achieved around 20% top-1 accuracy using cross-validation on the GNPS training set, but with a different choice of candidate set for each spectrum, and a better curated set of spectra. In our experiment, IOKR outperforms the baseline by a considerable margin, especially at low values of *n*, with an AUC (i.e. the area under the top-*n* curve for varying values of *n*) of 0.6534 compared to 0.5209 for the null distribution. In this context, the AUC can be considered as a measure of how close we are to the top hit being the correct one in all instances (AUC of 1.0) compared to a randomised baseline.

### 3.3 Evaluating the performance of IOKR

Similarly to the evaluation of the standardised strain correlation score, we can observe the distribution of the scores for the validated links among the scores for all potential links in the Crüsemann data set. Out of 3316 BGCs in the data set, 2242 could be assigned structure based on similarity to MIBiG entries, and used as candidate set for the 6246 MS2 spectra in the data set. As can be seen from Fig 6, the upper end of the distribution for the IOKR score contains a relatively high proportion of the validated links, with the mean score of 0.0105 for all links and 0.0364 for validated links (Table 1). Results for other data sets can be found in Table C in S1 Text, as well as in S2 and S3 Figs.

As an additional level of validation, we tested some high scoring links by exploring whether it was possible to manually match peaks in the MS2 spectra to the chemical structures. We then assessed how much these peaks influenced the ranking produced by IOKR. If peaks that can be matched to chemical structures have a large influence on ranking, it provides additional evidence to the validity of the links. We found several examples of this in both validated and prospective links, details of which are provided in Appendix D in S1 Text.

### 3.4 Complementarity of IOKR and strain correlation scoring

If multiple scoring methods are complementary then using them in concert makes sense. One way to demonstrate the efficacy of individual scores is to test whether or not the upper percentiles of their distributions are enriched with validated links, i.e. contain a relatively higher number of validated links compared to the data set as a whole. Table 2 shows the proportion of links that are validated across the whole set of links (*Total*; top row), and for links above the

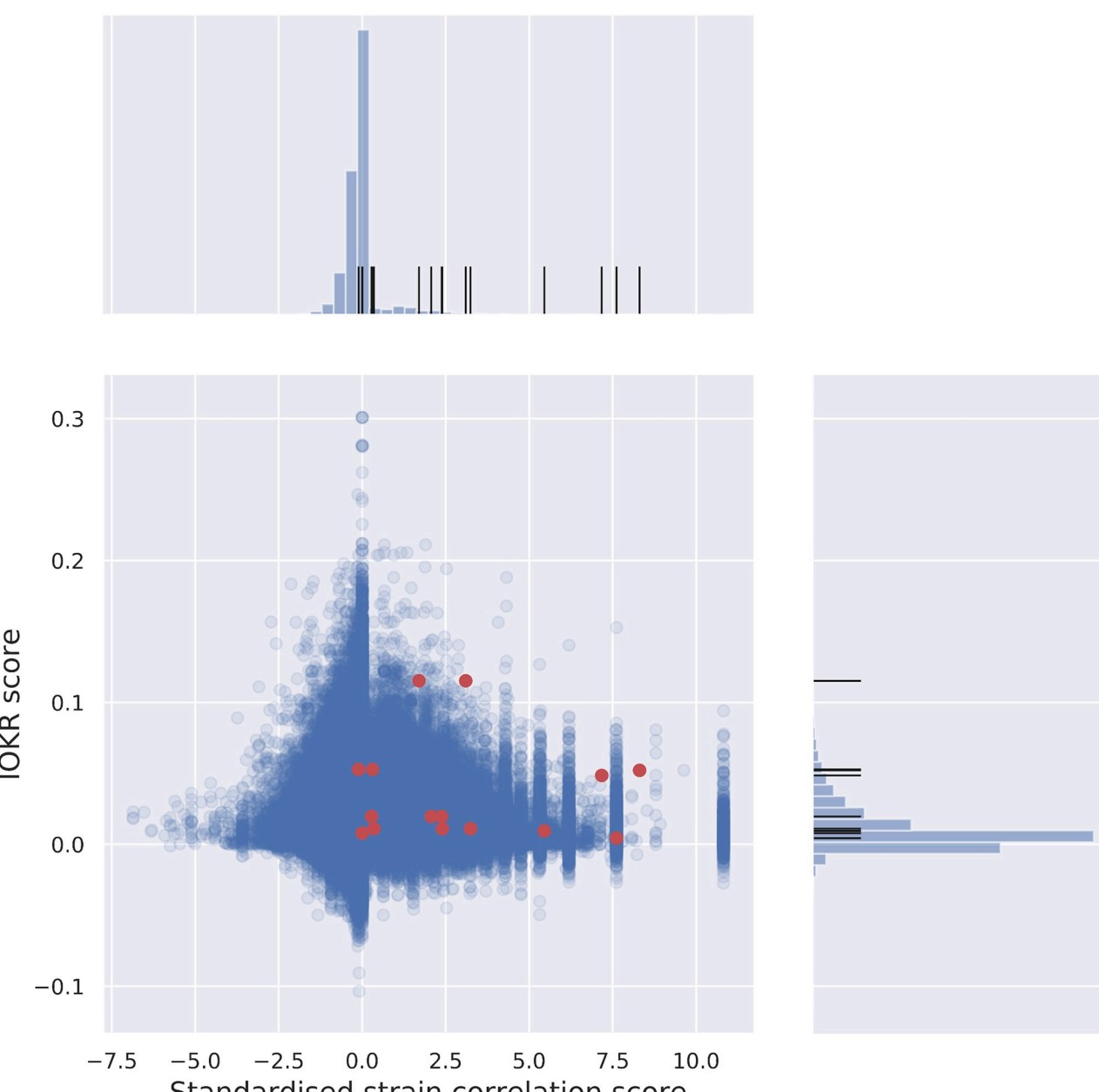

**Fig 6. Correlation of IOKR- and strain correlation scores.** IOKR- and strain correlation scores for all potential links in the Crüsemann data set, with histograms of the scores. Validated links are indicated in red on the joint plot, and with black lines on the distribution histograms. Validated links are concentrated in the upper-right quadrant, i.e. score relatively high on both axes. Figures for the two further data sets can be found in S3 Fig.

90th percentile for raw correlation, standardised correlation and IOKR scores (second, third and fourth rows resp.). Looking at Table 2, we can see that for all methods, the 90th percentile contains a higher proportion of validated links than across all links, and the standardised correlation score consistently improves upon the raw correlation score. Note that the rather low value of proportions throughout is largely due to the small number of validated links. The number of true hits in the dataset (and in the 90th percentiles) is likely to be much higher. The final row in Table 2 shows the proportions when looking above the 90th percentile for both

the IOKR and standardised scores together. This gives the highest proportions across all three datasets, demonstrating their complementarity, and hence the potential in combining the scores. The distribution of the scores for the Crüsemann data set, and the relative score of the validated links, can be seen in the histograms of Fig 6.

Since the number of validated links in each data set is small (ranging from 5 to 15), we can pool the links across the three datasets to get a clearer sense of the statistical significance. Considering the 90th percentile per data set for both scores, and adding up the numbers of links in each category (validated or unvalidated, and scoring above 90th percentile for either or both scores) for all the data sets, both the IOKR and standardised strain correlation scores are significantly enriched (*p*-value of 0.0139 and $2.483 \times 10^{-11}$, respectively) for validated links. Furthermore, the set of links scoring above the 90th percentile on both scores is significantly enriched compared to the set that exceed either of the individual scores (*p*-value of $2.633 \times 10^{-4}$ and 0.0208 starting from IOKR and standardised strain correlation scores, respectively). Relevant tables for 90th and 95th percentiles can be found in Table D in S1 Text, while the top-scoring links for each data set can be found in S2–S4 Data.

## 3.5 Scoring potential links for a particular BGC

A common approach to establishing correspondence between a GCF and a metabolite is to start with a GCF with established homology to a particular MIBiG BGC. The question then becomes one of ranking the potential GCF-MF links *for that particular GCF* to find the most likely product. Starting from a GCF, one can order the list of potential GCF-MF links by standardised strain correlation score, and use the IOKR score to further prioritise the ordered list for verification, or vice versa.

Starting with a GCF in the Crüsemann data set with homology to a particular MIBiG BGC, and a validated link to a MF, Table 4 shows how many out of the 3094 potential links involving that GCF score as high or higher than the validated link on the IOKR score (col. 2), the standardised strain correlation score (col. 3), or both scores (col.7). For instance, 1151 out of 3093 links including BGC0001228 (retimycin A, validated link excluded) have an equal or higher standardised strain correlation score than the correct link, of which 140 also have an equal or higher IOKR score. Similarly, 15 out of 3093 links containing BGC0000241 (lomaiviticin A, validated link excluded) have an equal or higher standardised strain correlation score, none of which has an equal or higher IOKR score than the correct link. The distributions of scores for a selection of the BGCs belonging to validated links can be seen in Fig 7.

Whilst this, and the previous section, demonstrates clearly that the scores are complementary, this information in itself does not help us to rank the potential links, since it requires knowledge of the score of the true link, which is not known a priori. One way of using both scores simultaneously to rank this list is to combine them into a single score. Optimising the exact combination of scores is outside the scope of this article. However, we can demonstrate the utility of doing so, with the understanding that further optimisation may be possible.

To make the IOKR score comparable to the standardised correlation score, we can standardise it in a similar manner to the strain correlation score. Letting $\sigma_{\text{IOKR}}$ be the IOKR score, we define the standardised IOKR score $\bar{\sigma}_{\text{IOKR}}$ as

$$\bar{\sigma}_{\text{IOKR}} = \frac{\sigma_{\text{IOKR}} - E[\sigma_{\text{IOKR}}]}{Var[\sigma_{\text{IOKR}}]}$$

where the variance and expected value are taken over the set of all potential links.

In order to define the function used to combine the two scores, we can consider a two-dimensional space where each axis corresponds to one of the scores, and concentric rings

**Table 4. Scoring function performance.**

| MIBiG ID | BGC | GCF | MF | Rank of validated link | | | | | |
|---|---|---|---|---|---|---|---|---|---|
| | | | | $\sigma_{\text{IOKR}}$ | $\bar{\sigma}_{\text{corr}}$ | $\ell_{\frac{1}{2}}$ | $\ell_1$ | $\ell_2$ | $\sigma_{\text{IOKR}}$ and $\bar{\sigma}_{\text{corr}}$ |
| BGC0001228 | 990 | 161 | 492 | 384 | 1152 | **253** | 271 | 317 | 141 |
| BGC0000241 | 2688 | 377 | 381 | 119 | 16 | **5** | 12 | 14 | 1 |
| BGC0000333 | 739 | 132 | 489 | 1810 | 25 | 32 | **12** | **12** | 12 |
| BGC0000827 | 1642 | 232 | 206 | 6 | **1** | 2 | **1** | **1** | 1 |
| BGC0001830 | 2104 | 295 | 309 | 673 | **7** | **7** | 9 | 24 | 4 |
| BGC0000137 | 394 | 71 | 353 | 80 | 16 | **2** | 14 | 49 | 1 |
| BGC0000137 | 394 | 333 | 353 | 78 | 189 | **12** | 29 | 66 | 3 |
| BGC0000137 | 394 | 71 | 358 | **798** | 1968 | 1232 | 997 | 920 | 534 |
| BGC0000137 | 394 | 333 | 358 | 636 | 612 | **409** | 642 | 735 | 120 |
| BGC0000137 | 711 | 48 | 353 | **6** | 287 | 36 | 214 | 261 | 1 |
| BGC0000137 | 711 | 123 | 353 | **6** | 126 | 14 | 105 | 122 | 1 |
| BGC0000137 | 711 | 367 | 353 | 6 | 216 | **4** | 85 | 203 | 1 |
| BGC0000137 | 711 | 48 | 358 | 184 | 282 | **146** | 271 | 285 | 17 |
| BGC0000137 | 711 | 123 | 358 | 184 | 101 | **28** | 92 | 99 | 7 |
| BGC0000137 | 711 | 367 | 358 | 184 | 90 | **19** | 89 | 88 | 7 |
| no. of best ranks | | | | 3 | 2 | **10** | 2 | 2 | |

The first four columns show the MIBiG ID and BGC, GCF and MF IDs of the validated links. Columns five and six show the rank of the validated link ordered by $\sigma_{\text{IOKR}}$ and $\bar{\sigma}_{\text{corr}}$. Columns seven through nine show the rank of the validated link using the $\ell_p$ scoring functions. The last column shows the rank of the validated link using both $\sigma_{\text{IOKR}}$ and $\bar{\sigma}_{\text{corr}}$, i.e. the ranking of the link in the product order. Lower numbers indicate better ranking, and the best rank (excluding the product order) for each validated link is indicated in bold. Multiple appearances of the same MIBiG IDs are due to multiple validated MFs for a single BGC, and antiSMASH mapping the same ID to multiple GCFs, as discussed earlier.

centered at the origin. In the quadrant where both scores are positive, quadrant I, the links further from the origin should be assigned better ranking. Looking at the set of all circles centered at the origin, we can then order the links by the radius of the circle on which they lie, with a preference for larger radius.

Looking at Fig 6, however, the scores extend along the axes (close to each score being 0), rather than forming a circle. Therefore, rather than restrict ourselves to computing distance from the origin wih the Euclidean norm, $\ell_2$, we can consider the more general $\ell_p$-norm where in the 2-dimensional case

$$\ell_p(x, y) = (|x|^p + |y|^p)^{\frac{1}{p}}, p \in \mathbb{N} \tag{1}$$

In our application, we can also consider values of $p$ such that $0 < p < 1$. Although in those cases $\ell_p$ is not a norm, since it does not fulfill the triangle inequality, this is not a problem when using it to rank combined scores. Fig 8 shows the set of points $(x, y)$ such that $\ell_p(x, y) = 1$ for three values of $p$, demonstrating the parallels of the circle in $\ell_{\frac{1}{2}}$ to the distribution of scores in Fig 6.

In the positive quadrant (I) of the coordinate system, we can use Eq 1 directly, with $x$ as the standardised correlation score and $y$ as the standardised IOKR score. To penalise scores in the other quadrants, particularly quadrant III, we can multiply the $x$- and $y$-terms with the corresponding sign. Using $\ell_{\frac{1}{2}}$, our combined scoring function is therefore

$$\sigma_{\text{sum}} = \text{sgn}(\bar{\sigma}_{\text{corr}})\sqrt{|\bar{\sigma}_{\text{corr}}|} + \text{sgn}(\bar{\sigma}_{\text{IOKR}})\sqrt{|\bar{\sigma}_{\text{IOKR}}|} \tag{2}$$

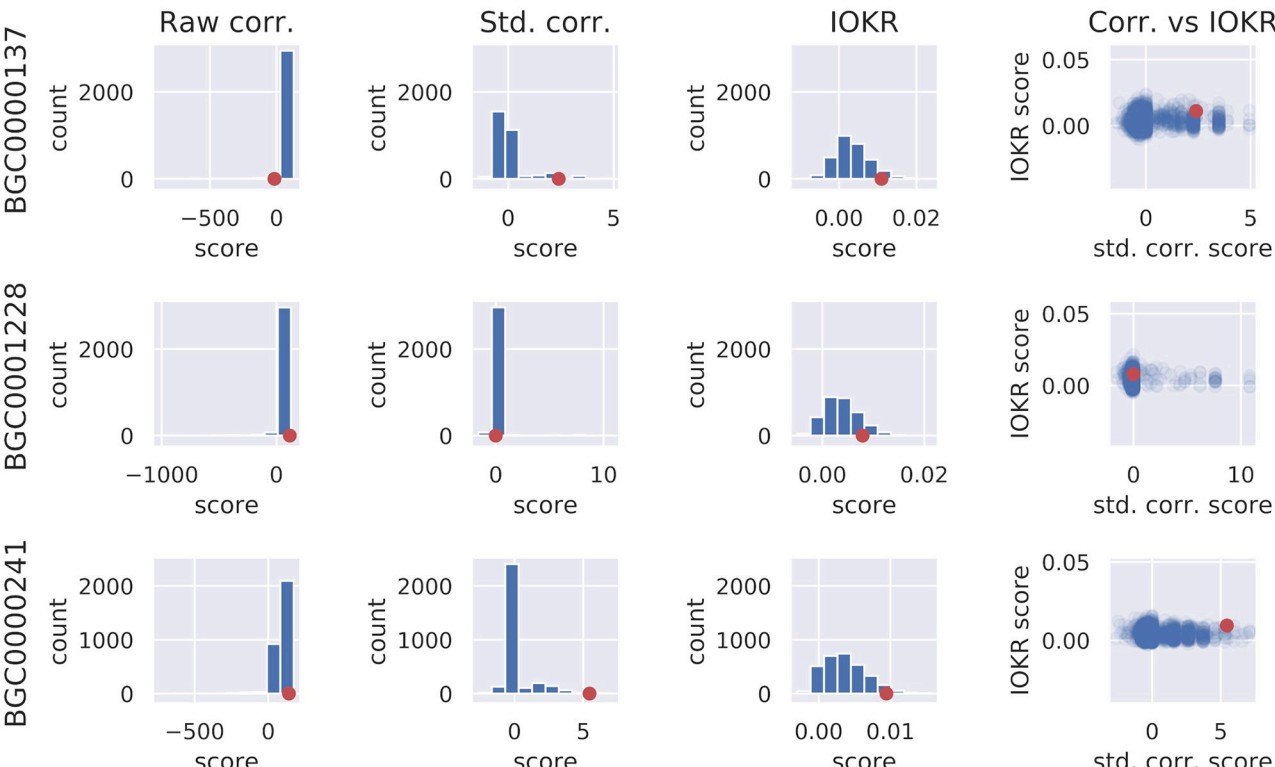

**Fig 7. Scores starting from particular GCF.** Position of the score for the validated GCF-MF pair (red) within the distribution of the scores of the links between that particular GCF and all MFs, for a selection of validated links in the Crüsemann data set (rows). The first three columns show histograms of the raw and standardised versions of the strain correlation score, as well as the IOKR score, for all links including a given GCF, with the score of the correct link indicated. The last column shows the standardised correlation score ($x$-axis) and IOKR score ($y$-axis) for the same links, again with the correct link indicated. Both IOKR and the standardised correlation scores tend to put validated links higher in the distribution of scores for the GCF in consideration, than the raw correlation score. Furthermore, some of the validated links score relatively higher on IOKR than the standardised strain correlation score, and vice versa, suggesting that the two scores complement one another. For full results, as well as for other data sets, please refer to S4 Fig.

where $\bar{\sigma}_{\mathrm{corr}}$ is the standardised correlation score and $\bar{\sigma}_{\mathrm{IOKR}}$ the standardised IOKR score, and sgn is the sign function mapping positive values to 1 and negative values to −1.

The rank of the validated links according to the various scoring functions can be seen in Table 4. The first two columns show the number of links scoring higher or equal to the validated link ordered by the IOKR and the standardised correlation scores, while the next three columns show the same where links have been ordered by the combined scores for $\ell_{\frac{1}{2}}$, $\ell_1$ and $\ell_2$. In 10 out of the 15 validated links considered, the $\ell_{\frac{1}{2}}$ score assigns the best rank to the validated link, including in three out of the first five cases where the link is unambiguous. For instance, the validated link for BGC0001228 (retimycin A) is ranked at number 253 and for BGC0000241 (lomaiviticin A) at number 5, both of which are considerably better than for either scoring function on their own, as well as for the other values of $p$ tested.

The last column of Table 4 shows the rank of the validated links in the product order. While in all cases resulting in the best ranking of the validated links, in practice the product order is not particularly useful as it yields a very high number of links that are not comparable using the relation, i.e. all links $a$, $b$ where $\sigma_{\mathrm{IOKR}}(a) < \sigma_{\mathrm{IOKR}}(b)$ but $\sigma_{\mathrm{corr}}(b) < \sigma_{\mathrm{corr}}(a)$, or vice versa.

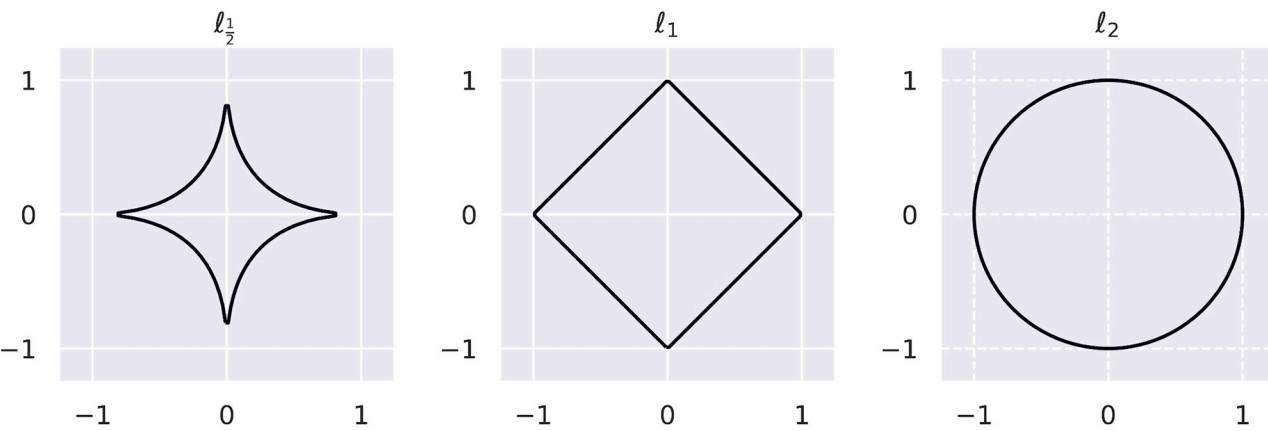

**Fig 8. Combining scores.** The set of points $(x, y)$ such that $\ell_p(x, y) = 1$, for three different values of $p$. This shows the form of the iso-lines of scores using the $\ell_p$ function for different values of $p$ to combine the scores.

While none of the scoring functions evaluated in Table 4 consistently ranks the validated links at the top of the set of potential links for the BGC in question, combinations of both scores, in particular $\ell_{\frac{1}{2}}$, tend to assign better rank to the validated links than the individual scoring functions. Other values of $p$, or other functions to combine the scores, may improve these results. However, we believe our analysis demonstrates the utility of using both the standardised strain correlation score and the IOKR score as complementary scoring functions to prioritise the list of potential links.

## 4 Discussion

We have shown that standardising the strain correlation score makes it more effective at prioritising validated links relative to all links, and introduced IOKR as a complementary feature-based scoring function. The major strength of IOKR compared to other feature-based methods is that it is not dependent on product type, but yields scores that are directly comparable between different product types. We have also demonstrated that considering both scoring methods at the same time increases the ratio of validated links among high-scoring links. Furthermore, by pairing the MIBiG and GNPS databases, and using the Paired omics Data Platform, we introduced data sets to test the efficiency of the scoring methods, both separately and combined.

The standardised strain correlation score still suffers from the drawback inherent in correlation-based scoring, of not being able to distinguish between potential links showing the same pattern of strain presence or absence. An obvious example is prioritising multiple singleton GCFs and MFs for the same strain. As a complementary scoring function to the strain correlation score, IOKR does not have this limitation. While it is not theoretically dependent on product type, due to insufficient test set size, breaking down performance by product type to verify this is currently difficult. A drawback of the current IOKR scoring method is its reliance on MIBiG homology to assign molecular structures to BGCs, which is needed to compute the molecular fingerprint. This restricts its use to those BGCs which show considerable homology with MIBiG entries. While still useful in this form, predicting molecular fingerprints directly from BGCs would broaden the applicability of the scoring function.

As stated earlier, IOKR is also highly dependent on the choice of both kernel function and molecular fingerprints. As the molecular fingerprints denote particular substructures of the

molecules, creating additional fingerprints that specifically target molecular substructures seen in secondary metabolites may improve performance. Similarly, the kernels used on the MS2 space can almost surely be further optimised, both through Multiple Kernel Learning [27] and other approaches such as vector embeddings [41].

## 5 Conclusion

Leveraging the correspondence between data from multiple microbial strains is an important tool to aid in linking BGCs to spectra. Standardising the strain correlation score by taking into account the number of strains involved in the link makes the score more useful by minimising the effect of strain count on the score, thereby no longer favouring common BGCs and molecules. We also introduced IOKR as a novel feature-based scoring method for potential BGC-spectrum links, and showed how the two methods complement one another by demonstrating the relative enrichment of previously validated BGC-spectrum links amongst the top-scoring links in both scoring functions. By using both scores simultaneously, the prioritisation of hypothetical links can be made more effective. Finally, we introduced the NPLinker framework to aid in prioritising BGC-spectrum links for further research. We believe that our work provides the natural products community with new tools that ease the combined analysis of genome and metabolome mining approaches. This may pave the way toward the discovery of novel chemistry that is much needed in diverse areas of human medicine and agriculture.

## 6 Supporting information

**S1 Text. Supplementary information for *ranking microbial metabolomic and genomic links in the NPLinker framework using complementary scoring functions*.**
(PDF)

**S1 Fig. Raw vs. standardised strain correlation scores.** Histograms showing the distribution of raw and standardised strain correlation scores for the microbial data sets, as well as positions of validated links within the distribution.
(PDF)

**S2 Fig. IOKR score of validated links.** Histograms of the distribution of IOKR scores for the microbial data sets, as well as positions of validated links within the distribution.
(PDF)

**S3 Fig. IOKR vs. correlation score.** Figures showing the IOKR- and standardised strain correlation scores for the microbial data sets, with the validated links marked.
(PDF)

**S4 Fig. Score distributions for a particular BGC.** Figures showing the distributions of scores starting from BGCs in validated links for the microbial data sets.
(PDF)

**S1 Data. Linked MIBiG and GNPS databases.** File containing GNPS id, MIBiG id and SMILES string, linking the two databases.
(ZIP)

**S2 Data. High-scoring links from the Crüsemann data set.**
(ZIP)

**S3 Data. High-scoring links from the Leão data set.**
(ZIP)

**S4 Data. High-scoring links from the Gross data set.**
(ZIP)

## Author Contributions

**Conceptualization:** Grímur Hjörleifsson Eldjárn, Justin J. J. van der Hooft, Katherine R. Duncan, Sylvia Soldatou, Simon Rogers.

**Data curation:** Grímur Hjörleifsson Eldjárn, Andrew Ramsay, Simon Rogers.

**Formal analysis:** Grímur Hjörleifsson Eldjárn, Simon Rogers.

**Funding acquisition:** Katherine R. Duncan, Simon Rogers.

**Investigation:** Grímur Hjörleifsson Eldjárn.

**Methodology:** Grímur Hjörleifsson Eldjárn, Juho Rousu, Simon Rogers.

**Project administration:** Simon Rogers.

**Software:** Grímur Hjörleifsson Eldjárn, Andrew Ramsay, Rónán Daly, Joe Wandy.

**Supervision:** Katherine R. Duncan, Juho Rousu, Simon Rogers.

**Validation:** Justin J. J. van der Hooft, Katherine R. Duncan, Sylvia Soldatou.

**Visualization:** Grímur Hjörleifsson Eldjárn, Andrew Ramsay.

**Writing – original draft:** Grímur Hjörleifsson Eldjárn.

**Writing – review & editing:** Grímur Hjörleifsson Eldjárn, Justin J. J. van der Hooft, Katherine R. Duncan, Sylvia Soldatou, Juho Rousu, Simon Rogers.

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
