## [Decision Letter · Decision Letter 0]

4 Jan 2021

Dear Dr Rogers,

Thank you very much for submitting your manuscript "Ranking microbial metabolomic and genomic links in the NPLinker framework using complementary scoring functions" for consideration at PLOS Computational Biology.

As with all papers reviewed by the journal, your manuscript was reviewed by members of the editorial board and by several independent reviewers. In light of the reviews (below this email), we would like to invite the resubmission of a significantly-revised version that takes into account the reviewers' comments.

All three reviewers found the manuscript to be of interest. Reviewers 1 and 2 have several concerns particularly related to validation of the methods described here which need to be addressed in a revised manuscript.

We cannot make any decision about publication until we have seen the revised manuscript and your response to the reviewers' comments. Your revised manuscript is also likely to be sent to reviewers for further evaluation.

Sincerely,

Niranjan Nagarajan

Associate Editor

PLOS Computational Biology

Jason Papin

Editor-in-Chief

PLOS Computational Biology

All three reviewers found the manuscript to be of interest. Reviewers 1 and 2 have several concerns particularly related to validation of the methods described here which need to be addressed in a revised manuscript.

Reviewer's Responses to Questions

**Comments to the Authors:**

Reviewer #1: see attached review document

Reviewer #2: The manuscript titled “Ranking microbial metabolomic and genomic links in the

NPLinker framework using complementary scoring functions” by Eldjárn et al demonstrates a ranking scheme to score link the genomic and the metabolomic data. Specifically, the authors use the information about strains, biosynthetic gene clusters (BGCs) and the metabolic fingerprints (MFs) obtained from mass spectra. The authors raise concerns over the existing scoring scheme and propose a method to modify this score to circumvent the challenges. Besides, the authors propose the use of an Input-Output Kernel Regression (IOKR) to predict the ranking of BGCs by combining the information from strains and MFs. Further, they integrate the scores obtained from the above two proposed methods and suggest that the combined score is better. Overall, I find the method reasonable; however, the proposed method requires further validations. Currently, the validation relies heavily on the predictions from antiSMASH, although there are other BCGs tools available. Further, at some places in the manuscript, I observe that the scores obtained using IOKR are lower, raising critical questions about the method proposed. I have listed my suggestions below.

Major

1. Lines 158-159 - What do the correlation scores indicate? What is the range of the correlation scores?

2. Lines 176-177 - For compounds having similar structures, how is the mapping done from MS2 spectra to the space of metabolites? In such a case of very similar compounds, how would the scores vary? How do the authors handle the mass spectra of cyclic compounds and stereoisomers?

3. Lines 212-213 – Since the input spectra are filtered to include only the peaks in training data, wouldn’t there be a bias?

4. It is quite unclear from the methods how the authors link the mass spectrum with molecular fingerprints. I suggest that the authors explain this further in the manuscript.

5. It is also not clear how the problems reported in lines 223-235 are alleviated by combining the two techniques since Table 1 does not report that scores obtained from IOKR+standardized correlation.

6. Table 1 – Scores of the links reported by IOKR offers very little improvement over the standardized score – in which case, what is the novelty behind using IOKR? I believe it is important for the authors to discuss this in the main manuscript.

7. Lines 372-376 is quite unclear – What is a correct metabolite? Suggest rephrasing for clarity.

8. Line 374 – How are the rank ties resolved?

9. Figure 7 – has missing axes labels and hence can’t be interpreted.

10. How do the links given by standardized correlation score and IOKR vary? Is there a degree of overlap?

11. Table 4 - Since there are so many links above the validated links, what does it mean? Based on the claims in Results section 1, I would imagine that validated link from the combination of two scores (l_1/2) to have the highest rank (but do not observe it in Table 4). Shouldn’t the validated links have a higher rank compared to the others? This should be discussed.

12. Also, since BGC0000137 has multiple entries, which rank should be prioritised and used for further analyses? This is an important point to be discussed in the manuscript.

13. Since the authors are proposing a new method, I would suggest that the comparison must be done with BGCs from other tools like DeepBGCs as well.

14. The link to the tool NPLinker is not provided in the main manuscript. Hence, this couldn't be used/tested.

Minor

1. At a few places such as Lines 452, the notations are not consistent

2. At several places these terms are used Established links, validated links, verified links and it becomes difficult to comprehend. I suggest authors use consistent terms or explain what each of these terms mean in the supplement.

3. Small typos need to be fixed (e.g. Summary prodcut -> product)

Reviewer #3: Abstract: Single paragraph is recommended.

There are several typos in the manuscript, “prodcut, unknnown, concenrated, compuational, peptidogenomcs, severly, to produces, “ etc.

There are some acronyms such as RiPP, but the open form is not given in the manuscript.

Table 2 is first mentioned in line 357, but without any details on its rows (what is total score etc). This information is provided much later, when Table 2 is mentioned again. Please either remove the first referral or move the related explanations to the place where Table 2 is first mentioned.

Crüsemann figure (Figure 6) is repeated in the supplementary file section 10. In the related supplementary figures for other datasets, please use red, not green, for verified points. Green is difficult to see.

Table 4: What do the bold numbers show? This should be added to table caption.

**Have all data underlying the figures and results presented in the manuscript been provided?**

Reviewer #1: Yes

Reviewer #2: Yes

Reviewer #3: Yes

PLOS authors have the option to publish the peer review history of their article (what does this mean?). If published, this will include your full peer review and any attached files.

Reviewer #1: No

Reviewer #2: No

Reviewer #3: **Yes: **Tunahan Cakir
---

## [Decision Letter · Decision Letter 1]

26 Mar 2021

Dear Dr Rogers,

We are pleased to inform you that your manuscript 'Ranking microbial metabolomic and genomic links in the NPLinker framework using complementary scoring functions' has been provisionally accepted for publication in PLOS Computational Biology.

Best regards,

Niranjan Nagarajan

Associate Editor

PLOS Computational Biology

Jason Papin

Editor-in-Chief

PLOS Computational Biology

Reviewer's Responses to Questions

**Comments to the Authors:**

Reviewer #1: The authors have carefully considered the comments and concerns of this reviewer, and responded in a very scholarly and appropriate fashion. The revised paper is significantly improved and will make a fine contribution to the field.

Previous Comment Number (also used in the Author response):

1. It appears that the authors have solved the issues.

2. The additions by the authors respond to and answer our suggestion that they try matching substructures to fragmentation spectra.

3. The addition of the clarification in section 2.6 is very good!

4. Based on their arguments, this reviewer agrees that the quality of the MAG-BGCs would not add much to the main conclusions of the paper.

5. The authors have provided a very good response to this issue.

6. The explanation provided by the authors appears valid.

7. This reviewer agrees that there are adequate comparisons against the current state-of-the-art, which is the strain correlation score defined by Doroghazi et al

8. The added text helps to explain the accuracy.

Items 9-19 – All of these issues have been properly dealt with by the authors, and good explanations provides.

Reviewer #2: Thank you for carefully considering all the comments and making appropriate modifications. The revised manuscript looks substantially stronger than the earlier submission.

Reviewer #3: My comments were properly addressed by the authors.

**Have all data underlying the figures and results presented in the manuscript been provided?**

Reviewer #1: Yes

Reviewer #2: Yes

Reviewer #3: Yes

PLOS authors have the option to publish the peer review history of their article (what does this mean?). If published, this will include your full peer review and any attached files.

Reviewer #1: No

Reviewer #2: No

Reviewer #3: No

---

## [Editor Report · Acceptance letter]

22 Apr 2021

PCOMPBIOL-D-20-01813R1 

Ranking microbial metabolomic and genomic links in the NPLinker framework using complementary scoring functions

Dear Dr Rogers,

I am pleased to inform you that your manuscript has been formally accepted for publication in PLOS Computational Biology. Your manuscript is now with our production department and you will be notified of the publication date in due course.

With kind regards,

Andrea Szabo
